# Effects of the FIFA 11+ and a modified warm-up programme on injury prevention and performance improvement among youth male football players

**Mojtaba Asgari** [1]*, **Mohammad Hossein Alizadeh**[2ᵒ], **Shahnaz Shahrbanian**[3ᵒ], **Kevin Nolte**[1ᵒ], **Thomas Jaitner**[1ᵒ]

**1** Movement and Performance Science Department, Institute for Sport and Sport Science, TU Dortmund University, Dortmund, Germany, **2** Sports Medicine Department, Faculty of Physical Education and Sports Science, University of Tehran, Tehran, Iran, **3** Department of Sport Science, Faculty of Humanities, Tarbiat Modares University, Tehran, Iran

ᵒ These authors contributed equally to this work.
* mojtaba.asgari@tu-dortmund.de

**Data Availability Statement:** *** PA @ ACCEPT: Please follow up with authors for data availabile at

## Abstract

### Introduction

The effects of the FIFA11+ programme (the 11+) on ankle and groin injuries and performance have remained questionable. The latter, particularly, has potentially reduced the implementation rate and applicability of the programme. This study aimed to evaluate the mid-to-long-term effects of the 11+ and a modified programme including football-specific exercises on injury prevention and performance improvement.

### Materials and methods

Three teams of the Iranian Youth League (division two) volunteered to participate in this study and were randomly assigned to two intervention groups (F11+; n = 29, M11+; n = 31) and a control group (n = 30). The F11+ followed the FIFA 11+ programme, whereas the M11+ performed modified exercises three times weekly as a warm-up protocol before training and competition through a football season. The control group carried out its routine warm-ups, including joggings, basic football drills, and static stretches, while having no injury prevention approaches. Lower extremity injuries, as well as exposure time for each player, were recorded. The football-specific performance was assessed using the Illinois Agility and Slalom Dribbling tests. ANOVA, Fisher Freeman Halton, and chi-square tests were used to analyze the data.

### Results

Injury incidences differed significantly between groups (p = 0.02, C = 0.40), with M11+ reporting the lowest incidence. Significant differences between the pre- and posttest as well as differences between the groups for development over time were revealed for the Illinois agility and dribbling speed (p≤0.01). Both performance tests demonstrated a large time

Accept *** We note that your Data Availability
Statement reads: "The dataset is uploaded on
Kaggle, a public repository platform for placing
data sets. URL:www.kaggle.com/dataset/

effect, as the effect sizes for time in agility and dribbling speed were 0.74 (CI = [0.66; 0.79]) and 0.86 (CI = [0.79; 0.87]), respectively. The effect size for the interaction can be categorized as medium, with 0.38 (CI = [0.25; 0.49]) for agility and 0.52 (CI = [0.40; 0.61]) for dribbling speed. M11+ showed the largest improvement in both.

## Discussion

Mid-to-long-term application of a structured dynamic warm-up that integrates injury prevention and performance approaches may lower injury incidences and improve youth subelite players' performance. Although additional studies with larger samples are needed to prove the results of the current study, the amateur clubs/teams could integrate such twofold dynamic warm up into their routine training plan and benefit its advantages on injury prevention and performance improvement.

## Introduction

Football accounts for nearly 300 million participants and is labeled the most popular but one of the riskiest Olympic sports [1, 2]. Missing training with a consecutive reduction in performance, dropout of young talent, early retirement, high socioeconomic costs, and increased risk of osteoarthritis are known as the consequences of sustaining injury [3–5]. Hence, mitigating football-related injuries has turned into the main focus of scientific inquiries. The FIFA 11+ programme (the 11+) was launched in 2006, aiming to integrate injury risk-mitigating exercises into a warm-up protocol. Given the advantages it entails in preventing injuries, it has been widely investigated and named a well-established injury prevention programme (IPP). The 11+ consists of three parts with an overall duration of 25 minutes and three difficulty levels. It aims to tackle the key elements of effective injury prevention programmes, such as neuromuscular control, static and dynamic balance, and the hamstring/quadriceps strength ratio [6].

The available literature strongly supports that the 11+ mitigates overall, thigh, and knee injuries among amature/subelite players [6–9]. More specifically, in men's subelite football, it reduces injury burden by up to 40% [10]. However, the efficacy of the 11+ in the prevention of ankle and groin injuries [6, 7, 11] and players' performance [12] has remained questionable. The latter, in addition to barriers such as prolonged duration, concerns regarding some exercises, and lack of a link to football-specific targets [13, 14], has been discussed as a potential reason for low compliance and less application of the programme [14]. Although high compliance has shown a strong correlation with the success of the IPPs and team success [15], O'Brien et al. (2017) reported that the 11+ was only implemented in 12% of the training sessions, while the modified forms were implemented in 28% of the sessions [14]. The most frequent reasons trainers gave for the modifications were adding variation, progression, challenge, and individualization [14, 16]. Additionally, Veith et al. suggested that part two of the FIFA11+ can be used outside of the training sessions since the trainers do not want to invest in it [17].

Although ankle injuries are among the most prevalent injuries in football [18–20], lower effects of the 11+ on such injuries have been observed [6, 7]. One possible explanation for this lack of efficacy could be the low volume of dynamic exercises in the first part, followed by extensive static exercises in the second part of the programme. Nonprogressive and static exercises may not stimulate control mechanisms of the postural system and therefore may not

reduce the risk of lower extremity injuries [21]. Studies by Veith et al. (2021) and Lopes et al. (2020) would precisely support this assumption, where they found no improvement in ankle stability, proprioception, and ankle evertors time latency following performing part two and the whole programme, respectively [17, 22, 23]. Hence, rescheduling the 11+ has been taken into account and appears to mitigate severe injuries and increase compliance [14].

Further, a lack of clarity exists regarding the effects of the 11+ on technical abilities and physical performance [12, 24–27]. Although it is widely accepted that an appropriate warm-up programme should improve performance [28], more recent studies indicated that the 11+ programme does not improve players' performance acutely but may reduce sprinting and agility compared to a dynamic warm-up programme. Another additional experiment demonstrated that the 11+ has no impact on kicking accuracy [29, 30].

Taken together, the main challenges regarding the 11+ correspond to low compliance and implementation rates, lower effects on ankle and groin injuries, and unclear effects on players' performance. Considering that the 11+ is a verified warm-up, a new framework is needed to incorporate different training components into this programme. Thus, modernizing and modifying the 11+ based on football-specific demands may properly tackle these challenges and become essential, as it has not been updated since its launch (2006). Therefore, this study aims to evaluate the effects of the 11+ (F11+) and a modified programme (M11+), including more football-specific exercises on injury prevention and performance improvement. We expect that (i) players in the F11+ and M11+ groups suffer fewer lower extremity injuries than players in the control group, (ii) M11+ results in a lower incidence of lower extremity injuries than F11+, and (iii) M11+ improves performance, while F11+ does not affect it.

## Methods

### Study design

This prospective study involved two interventions and a control group (CG). The F11+ and M11+ groups performed the 11+ and the modified 11+, respectively, three times weekly for warming up before training sessions. The CG continued its routine warm-up, including basic football drills, running, cutting movements, and static stretch without any specific injury prevention approaches. The intervention lasted four months (June to October), corresponding to a complete season, as only a few teams attended the league. At baseline, players performed agility and slalom dribbling tests as described below; the tests were then repeated by the end of the season before the last match. According to the consensus statement on injury definitions and data collection in prospective studies, injuries and exposure time were documented during the study period [32]. The ethics committee of the Sport Sciences Research Institute of Iran approved the study design and research methods (ID IR.SSRI.REC.1397.257).

### Participants and inclusion criteria

To calculate the sample size, we set the G-power Software version 3.1.9.4 (Universität Düsseldorf, Germany) as the exact test family, 2 tails, power of 80, alpha error 0.05, proportion 1 = 0.5, and proportion 2 = 0.15. The results revealed that 31 players per group are needed. Nevertheless, we asked all seven football clubs participating in Iranian youth league division three (northwest subdivision) to attend the study. The inclusion criteria were as follows: (i) competition in youth league division two, (ii) age range of 16 to 19 years at baseline, and (iii) at least two training sessions plus a match per week. Players were excluded if they (i) suffered severe lower extremity injury and trauma (exposure time ≥28 days) four months before the study, (ii) had a history of any musculoskeletal surgery, head concussion, or severe cardiovascular problems six months before the study, (iii) participated in other sports simultaneously or

(iv) missed over 10% of the training sessions. Teams were also excluded if they were already applying an IPP as a part of their daily routine training, which resulted in one exclusion. Consequently, only four teams (124 players altogether) were eligible to attend the study, one of which (26 players) declined the invitation. As a result, three teams involving 98 players agreed to attend the study, eight of which were then excluded based on the exclusion criteria. The 90 volunteered participants signed a written consent letter and were free to leave the study at any time. Further, using an automat block randomized algorithm, three blocks were created (block1 = F11+, block2 = M11+, and block three = CG), and each team was randomly assigned to a block. Before the study commencement, trainers and players of the F11+ and M11+ were instructed to become familiar with their specific IPP in separate sessions. Further, they received adequate supportive materials (the video clips and manual of the 11+ and photoslides of the M11+ exercises).

## Data collection and supervision

Medical assistants (licensed by the Sport-Medicine Federation of Iran) were responsible for diagnosing and documenting injuries. We defined an injury as "any complaint that results in a player being unable to take a full part in future football training or match play" [31]. Furthermore, we instructed trainers to record the exposure time using particular forms. While blinding each other and the study outcomes, two assistants were responsible for visiting the training sessions randomly and checking the accurate application of the interventions. Assessing compliance with the interventions was carried out by a threefold descriptive rating scale based on the percentage of fully completed warm-up sessions (HIGH>67% MOD<66%/LOW<33%) according to self-report data of trainers of the intervention groups.

## Performance assessment

We recruited Illinois Agility (IA), as well as Slalom Dribbling (SD), tests to evaluate Football-specific performance [32, 33]. Within-subject and between-subject differences in pre vs post-test were considered as the interventions' effects on those parameters. The IA performs in a rectangle of 10×5 m. The players start in the prone position, run toward the barrier at 10 m at maximum speed, return, and perform a zigzag run around four barriers, each 3.3 m apart. The test ends with another straight run to the end of the rectangle. For the SD, players dribble the ball after a starting signal in a zigzag shape around seven poles with a distance of one meter each and return a straight path of 10 meters. Each subject performed the test three times, and the best time, recorded through time gates, was considered for analysis.

## The modified programme

M11+ consists of four parts (seven, four, four, and five minutes) and follows the structure of F11+. The first part includes core exercises (prone, supine, and dynamic plank), balance exercises, and M. iliopsoas strengthening. The second part involves straight running, controlled contacts, hip in and hip out (with higher frequency than the F11+), a dynamic stretch of hamstring and rotational, and cutting movements. The third part consists of Nordic Hamstring Exercise, single-leg standing with a heel-to-toe movement cycle (to optimize the stimulation of mechanoreceptors of the ankle), and throwing a ball to the partner simultaneously as well as squats (lunges, squat and single-leg squat). The last part includes agility and plyometric exercises, drop jumps, and countermovement jumps, as proposed by Thomas et al. [34]. The exercises are categorized into three difficulty levels and would adapt every 2–3 weeks depending on the players' abilities and progress. By this, players may practice in a consecutive chain of static, dynamic, and intensive exercises and therefore should be ready for the main training body.

## Data analysis and statistics

Injury incidence and rate ratios of total lower extremity injuries and thigh, knee, and ankle injuries served as primary outcomes. We calculated injury incidence based on the overall number of injuries per group divided by time exposure in players' total hours of competition and training. Differences between pre- and posttest measures of IA and SD served as the secondary outcome. Chi-square and Fisher's Freeman Halton tests were used to analyze the injury incidences and rate ratios of the three groups [35]. If the chi-square test was applicable, the corrected contingency coefficients (C) were calculated. For the performance parameters, Kolmogorov-Smirnoff and Leven's tests were applied to verify the normality of the data and between-group homogeneity of variance, respectively. Repeated measures ANOVA analyzed differences between the interventions and the control group and the effect of the training period. The corresponding effect sizes (ESs) were determined by calculating $\eta^2$ and its 90% confidence interval (CI). Data were analyzed using SPSS software version 25, and the level of significance for all variables was set at $p \leq 0.05$.

## Results

Over the football season, 6381 hours of training and matches were recorded, indicating that each player participated in approximately 56 hours of training (1675.66 ± 229.32 min) and 15 hours of matches (451.33 ± 92.64 min). Overall, 31 lower extremity injuries affected 24 players (26.6%). ANOVA revealed no differences between groups in age (p = 0.18), height (p = 0.07), weight (p = 0.11) or BMI (p = 0.06), as presented in detail in Table 1. The injury incidences of the lower extremities for the M11+, F11+, and CG were 2.00, 5.79, and 7.55 per 1000 h, respectively, demonstrating that compared to the M11+, players of the F11+ had 2.89 times higher lower extremities. The risk ratios for the CG were 3.77 and 1.30 times higher than those for the M11+ and F11+ groups, respectively. The chi-square test derived a significant dependence for the overall number of injuries and the prevention programmes (p = 0.02, C = 0.40), illustrating the lowest value for the M11+ and the highest for the CG. The follow-up chi-square analysis produced no significant difference in the number of injuries between the F11+ and CG (p = 0.50, C = 0.12).

Separately, for thigh (p = 0.70), knee (p = 0.10), and ankle (p = 0.09) injuries, no dependency on the interventions was found. Compared to the F11+ group, the M11+ group revealed on average 2.33 times fewer thigh, knee, and ankle injuries, including no injury incidences for the knee and ankle (Table 2).

Fig 1 illustrates the pre- and posttest results of the three groups. All three groups started on a similar level, as the group means were within 0.1 s in dribbling speed and agility during the pretest. All three groups improved during the training period. Agility and dribbling speed displayed the same hierarchy in improvement and differences between the three groups. The M11+ group showed the most considerable improvements in group means, with 1.09 s and

**Table 1. Anthropometric data of all players (mean and std dev).**

| Parameter | M11+ | F11+ | CG | Sig |
|---|---|---|---|---|
| Age (year) | 16.98±0.7 | 16.96±0.68 | 16.69±0.58 | 0.18 |
| Height (cm) | 176±7.06 | 171.24±8.73 | 173.13±8.36 | 0.07 |
| Weight (kg) | 64.35±8.53 | 62.03±8.00 | 59.83±9.19 | 0.11 |
| BMI (kg/m$^2$) | 20.70±1.69 | 21.06±1.08 | 19.85±1.79 | 0.06 |

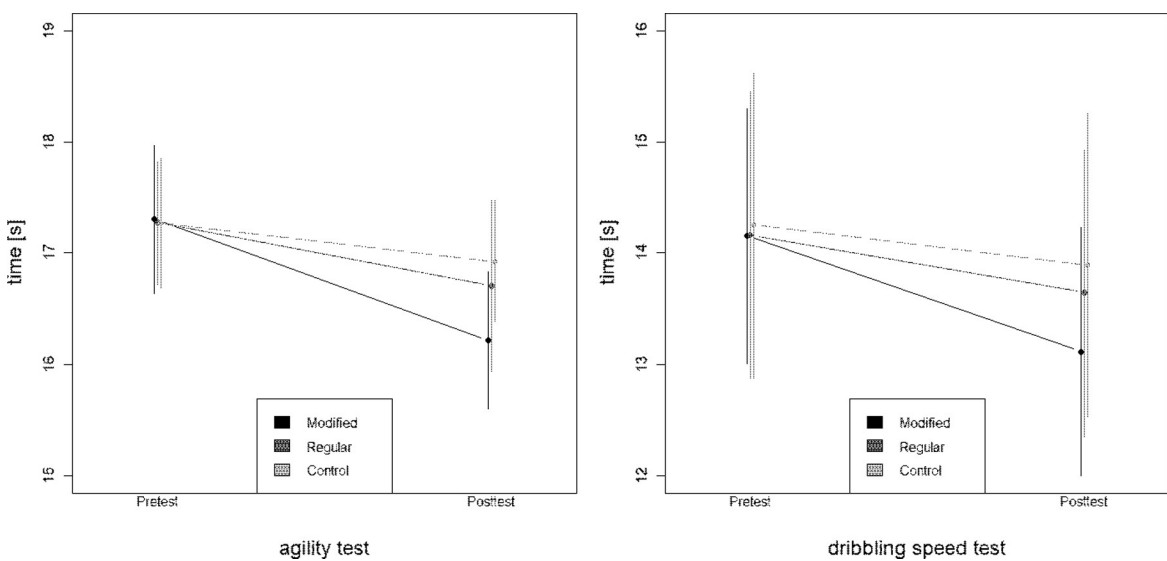

**Fig 1. Comparison pretest post-test in agility and dribbling speed.**

1.04 s, followed by the F11+ group, with 0.57 s and 0.52 s for agility and dribbling speed, respectively. The average times in the CG decreased by 0.35 s and 0.36 s, respectively.

A repeated-measure ANOVA supported these findings, as the effect of time and the interaction effect of group and time were significant ($p \leq 0.01$) for agility and dribbling speed. Both tests reveal a large time effect, as the effect size for time in agility is 0.74 (CI = [0.66;0.79]) and 0.86 (CI = [0.79;0.87]) in dribbling speed. The effect size for the interaction can be categorized as medium, with 0.38 (CI = [0.25;0.49]) for agility and 0.52 (CI = [0.40;0.61]) for dribbling speed (Table 3).

## Discussion

This study aimed to compare the effects of the FIFA11+ and a modified warm-up programme, including dynamic and football-specific exercises, on the prevention of injuries and performance improvement among young subelite male players. Overall, players who performed the F11+ as a warm-up routine suffered nearly 27% fewer lower extremity injuries than those who performed their routine warm-up, including basic football drills, running and cutting movements, and static stretch. However, this difference was not statistically significant, which conflicts with the previous studies of Soligard et al. (2008), Owoeye et al. (2014), and Granelli et al. (2015) but supports the study of Hammes et al. (2015), which observed similar results in

**Table 2. Lower extremity, thigh, knee, and ankle injuries.**

| Variable | | Total (n) | Control (n) | M11+ (n) | F11+ (n) |
|---|---|---|---|---|---|
| Lower extremity (iir) | Injured | 31 (4.86) | 15 (7.55) | 5 (2.0) | 11 (5.79) |
| | Non-injured | 59 | 15 | 26 | 18 |
| Thigh | Injured | 13 | 5 | 3 | 5 |
| | Non-injured | 77 | 25 | 28 | 24 |
| Knee | Injured | 6 | 4 | 0 | 2 |
| | Non-injured | 84 | 26 | 31 | 27 |
| Ankle | Injured | 7 | 4 | 0 | 3 |
| | Non-injured | 83 | 26 | 31 | 26 |

*iir = injury incidence ratio

**Table 3. Descriptive values and ANOVA results.**

| | Agility [s] | | Dribbling speed [s] | |
|---|---|---|---|---|
| Descriptive | Pretest | Posttest | Pretest | Posttest |
| Modified 11+ | 17.30 ± 0.66 | 16.21 ± 0.61 | 14.15 ± 1.15 | 13.11 ± 1.12 |
| Regular | 17.27 ± 0.55 | 16.70 ± 0.77 | 14.16 ± 1.29 | 13.64 ± 1.29 |
| Control | 17.27 ± 0.58 | 16.92 ± 0.54 | 14.25 ± 1.38 | 13.89 ± 1.37 |
| ANOVA | p-value | | p-value | |
| Group | 0.082 | | 0.396 | |
| Time | <0.001 | | <0.001 | |
| time*group | <0.001 | | <0.001 | |

veteran players [6, 9, 36, 37]. A possible explanation for such a nonsignificant difference could be the small sample size, short monitoring period, and consequently lower number of lower extremity injuries over the study period. Furthermore, many injuries in football occur during matches. However, our study participants were only involved in 15 hours of matches during a complete football season and were at lower risk of sustaining injury. Additionally, this study focused on lower extremity injuries, while other studies considered full-body injuries and reported significant differences.

On the other hand, modernizing and modifying the F11+ according to football-specific demands could optimize its effect on lower extremity injuries and improve the performance of young male football players. Players who performed the M11+ suffered significantly lower extremity injuries (up to 55%) than both the F11+ and control groups (p≤0.05). Along this line, Al Attar et al. (2017) showed that combining the 11+ with highly intensive posttraining exercises improves the programme's efficacy in preventing injuries [8]. In addition, the latest review on the literature pertaining to injury prevention strategies reported that football-related injuries can be mitigated by participating in dynamic warm-up programmes, including preventive exercises and adding strength, balance, and mobility exercises to the training sessions [38].

More specifically, our study revealed minor effects of F11+ on the prevention of ankle injuries, in line with the available literature. Lope et al. (2019 & 2020) reported that the 11+ produces no impacts on improving ankle stability and ankle evertor latency time [22, 23]. The dynamic stability of the ankle is mainly provided through the activity of the calf muscles. Additionally, most ankle injuries occur in the inversion position where the least joint stability exists, and the ankle is prone to injury. Therefore, delay in activation of ankle evertors that resist inversion and draw the ankle back to the upright position would reduce joint stability and put the athletes at higher risk of sustaining ankle injuries. Therefore, integrating plyometric, perturbation, and dynamic exercises that may properly stimulate proprioception into the warm-up strategy could be more beneficial in preventing ankle injuries. In the F11+ programme, there are more static than dynamic exercises, particularly in part two. The static structure of the F11+ could not stimulate feedback and control mechanisms of the postural system [22]. In contrast, dynamic exercises may adequately optimize the neuromuscular system and reduce severe ankle and knee injuries by improving proprioception [39]. Interestingly, the current study precisely addressed this assumption, where no ankle or knee injuries were observed in the M11+ group, albeit the differences were not statistically significant. Thus, further studies with larger sample sizes and better monitoring periods are required to prove our outcomes.

Neither F11+ nor M11+ could significantly reduce the risk of groin and hip injuries. Such lack of efficacy can be explained by the fact that no specific exercise for groin muscles is included in these warm-up modalities. This is an alarming deficiency and needs to be properly

addressed, as up to 12% of subelite football injuries occur in the groin region [40]. Harøy et al. suggested that using Copenhagen adductor exercises may fill this gap by providing missing eccentric hip adduction strength [41]. The Copenhagen exercises appear to be highly adaptable to the 11+, as they require no specific equipment and have three levels of difficulties [41]. The better efficacy of the M11+ in preventing hip injuries, however, might be due to increasing the volume of hip dynamic stretch exercise, namely, hip-in, in addition to alteration of core area exercises.

According to agility and dibbling performance, the pretest results did not show significant differences between the groups, meaning that the groups were averaged at the same level of expertise before the intervention. Post test results at the end of the intervention course revealed a significant difference between the groups. The M11+ group showed better performance in agility and dribbling speed tests (p≤0.05). There were no significant differences between the FIFA 11+ and CG (p≥0.05). From the performance standpoint, the present study supports the studies of Ayala et al. [30], Parsons et al. [24] and Dunsky et al. [29], which also addressed the lack of performance improvement followed by performing the 11+ programme but are in contrast to the studies of Bizzini et al. (2013), Zarei et al., and Hwang et al. [26, 27, 42]. Players of the M11+ group performed the agility test significantly better than the F11+ and control groups. However, differences in dribbling speed were only significant between M11+ and F11 +. Along with this, Daneshjoo et al. (2013) illustrated that F11+ might not improve the dribbling technique among youth players. Ayala et al., through their investigation of three different neuromuscular warm-up programmes, highlighted that F11+ would acutely decrease player sprinting compared to dynamic warm-ups [30]. Parameters such as agility and sprinting depend on the training features [43], so using exercises similar to real football's movement patterns/tasks may enhance these parameters. Most likely, the negative effect of F11+ on agility and dribbling is due to the static structure of the programme. The volume of the plyometric and anaerobic exercises in the M11+ players was twice as high as that in the F11+ players, which could be the main reason for the improved agility and dribbling speed among the M11 + players. In this regard, Thomas et al. (2009) indicated that plyometric exercises could improve football players' muscular power and agility [34]. As a result, implementing highly dynamic exercises as a part of warm-up routines but at a controlled intensity may be useful for the performance improvement of youth players.

As compliance with both programmes was calculated by MOD to HIGH, the study was not powered by compliance. However, another potential benefit of the M11+ might be increasing the compliance and implementation rate. According to the trainers' self-reports, the completion rate of the M11+ programme was slightly better than that of F11+ (66% vs. 61%). This statistic, parallel with the study of O'Brien et al. (2016), indicates that both trainers and players would prefer to implement dynamic warm-up protocols to be well prepared for the further intensive and competitive parts of the training and competition [44].

## Limitations

This study lacks a large sample size. Although we aimed to provide the most significant sample, three potential teams failed to meet the inclusion criteria and could not attend the study, and another team declined the invitation. This, alongside the short monitoring period caused by the small number of teams participating in the league, noticeably restricted us to manage a randomized controlled trial and somehow powered the outcomes.

## Conclusion

Incorporating performance-specific and injury prevention-specific exercises into a structured warm-up programme provides a twofold warm-up modality, which may not only mitigate injuries but also improve the performance of youth subelite football players simultaneously. Therefore, it would be practical for daily routine training and competitions and, as a result, can be better accepted and implemented by football administrations. The amateur clubs/teams are highly recommended to integrate such warm up into their routine training plan.

## Supporting information

**S1 Data.**
(XLSX)

## Acknowledgments

The authors would like to express their gratitude to Sahar Soheili for her comprehensive support in the data gathering and analysis phases.

## Author Contributions

**Data curation:** Mojtaba Asgari.

**Formal analysis:** Shahnaz Shahrbanian.

**Investigation:** Mohammad Hossein Alizadeh, Kevin Nolte.

**Methodology:** Mohammad Hossein Alizadeh, Thomas Jaitner.

**Project administration:** Mojtaba Asgari.

**Supervision:** Thomas Jaitner.

**Writing – original draft:** Mojtaba Asgari.

**Writing – review & editing:** Kevin Nolte.

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
