## [Decision Letter · Decision Letter 0]

6 Jul 2022

PONE-D-22-14375Effects of the FIFA 11+ and a modified warm-up programme on injury prevention and performance improvement among youth male football players.PLOS ONE

Dear Dr. Asgari,

Thank you for submitting your manuscript to PLOS ONE. After careful consideration, we feel that it has merit but does not fully meet PLOS ONE’s publication criteria as it currently stands. Therefore, we invite you to submit a revised version of the manuscript that addresses the points raised during the review process.

We look forward to receiving your revised manuscript.

Kind regards,

Rafael Franco Soares Oliveira

Academic Editor

PLOS ONE

Journal Requirements:

 [this study was not funded]. 

5. We note you have included a table to which you do not refer in the text of your manuscript. Please ensure that you refer to Tables 2 and 3 in your text; if accepted, production will need this reference to link the reader to the Table.

6. Please include your tables as part of your main manuscript and remove the individual files. Please note that supplementary tables (should remain/ be uploaded) as separate "supporting information" files.

7.  Please note that PLOS utilizes the CC BY 4.0 license (https://creativecommons.org/licenses/by/4.0/) which means that all material on our website is freely available online, and any third party is permitted to access, download, copy, distribute, and use these materials in any way, even commercially, with proper attribution. If the figures have already been published, authors must provide proper attribution, referencing the source clearly, and obtain permissions if the image is copyrighted.

Viewing your abstract, we can see that it was previously published by Springer:

"Al Attar, W.S.A., Soomro, N., Pappas, E. et al. How Effective are F-MARC Injury Prevention Programs for Soccer Players? A Systematic Review and Meta-Analysis. Sports Med 46, 205–217 (2016). " ext-link-type="uri" xlink:type="simple">https://doi.org/10.1007/s40279-015-0404-x"

Viewing this article at the Springer website, we see that all content and materials are copyrighted and are not available for re-use without permission.

To seek permission from the Springer to publish this Abstract under the Creative Commons Attribution (CC BY) 4.0 License, please contact them with the following text and PLOS ONE Request for Permission form (http://journals.plos.org/plosone/s/file?id=7c09/content-permission-form.pdf):

“I request permission for the open-access journal PLOS ONE to publish XXX under the Creative Commons Attribution License (CCAL) CC BY 4.0 (http://creativecommons.org/licenses/by/4.0/). Please be aware that this license allows unrestricted use and distribution, even commercially, by third parties. Please reply and provide explicit written permission to publish XXX under a CC BY license.”

Please upload the granted permission to the manuscript as a Supporting Information file.

Please note that RightsLink permission forms often impose use restrictions that are incompatible with our CC BY 4.0 license, and we are therefore unable to accept these permissions. For this reason, we ask that you contact the copyright holders with the PLOS ONE Request for Permission form.

If you are unable to obtain permission from the original copyright holder, please either remove the Abstract or supply a replacement Abstract that complies with the CC BY 4.0 license.

Additional Editor Comments:

Dear authors,

The work is interesting but some improvements can be made before further consideration. There are some sentence that lack support (references). Some limitations should be added/highlighted as well as some pratical applications for coaches.

Thank you

Reviewers' comments:

Reviewer's Responses to Questions

**Comments to the Author**

1. Is the manuscript technically sound, and do the data support the conclusions?

Reviewer #1: Yes

Reviewer #2: Partly

2. Has the statistical analysis been performed appropriately and rigorously? 

Reviewer #1: Yes

Reviewer #2: I Don't Know

3. Have the authors made all data underlying the findings in their manuscript fully available?

Reviewer #1: Yes

Reviewer #2: No

4. Is the manuscript presented in an intelligible fashion and written in standard English?

Reviewer #1: Yes

Reviewer #2: Yes

5. Review Comments to the Author

Reviewer #1: Dear authors,

I believe they did a good job and deserve credit for that.

The study is well designed, the introduction is robust, the methods are well explained and the discussion supports the results.

However, the conclusion can and should be improved, and should be presented with practical applications for the coach.

Furthermore, I believe that the limitations should be inserted at the end of the discussion.

Reviewer #2: The proposed manuscript is very interesting and has an noteworthy objective. Congratulations to the authors.

My suggestions follow below.

P.8

Comment 1:

Please rephase the “Background”, specifically “…the latter has potentially reduced compliance and

implementation of the programme”

Comment 2:

Keywords: I suggest not repeating the keyword that already exists in the title to enhance search results.

Comment 3:

“static stretches” in the warm-up? Or did the authors mean “dynamic stretches”?

Comment 4:

Is the discussion the conclusion in the abstract?

P.9

Comment 4:

I suggest rephrasing the sentence “It aims to tackle modifiable injury risk factors such as

neuromuscular control, static and dynamic balance, and the hamstring/quadriceps strength

ratio[8].” as they are not risk factors. They can be considred as “key elements of effective injury prevention programmes…”

Comment 5:

Please rephrase the sentence “The latter, along with barriers such as prolonged duration, concerns regarding some exercises, and lack of a link to football-specific targets[14, 18], has been discussed as a potential reason for low compliance and poor implementation of the programme[14, 19].”

Comment 6:

Please state the reference for the sentence “Although it is widely accepted that an appropriate warm-up programme should improve performance, more recent studies indicated that the 11+ programme does not improve players' performance acutely but may reduce sprinting and agility compared to a dynamic warm-up programme.”

P.10

Comment 7:

In the abstract the authors stated that the CG did not perform injury prevention studies, however in the methods section, the contrary is stated.

Comment 7:

Was static stretching really used?

P.11

Comment 8:

Please add further information of how the instructions to perform the 11+ and the M11+ were carried out.

Comment 9:

Data collection and supervision: How where the assistants “blinding each other”?

Comment 10:

Why was the “M. iliopsoas strengthening” and the “single-leg standing with a heel-to-toe movement cycle” selected? Can the authors support these choices with evidence? It would be great as the M11+ group showed few injuries.

P.12

Comment 11:

A table with Injury incidence/1000h and rate ratios for each group would enhance the manuscript.

Comment 12:

Further detail on injuries would also be of great interest for the study: (Fuller et al, 2006).

p.13

Comment 13:

The second manuscript from (Lopes et al.) is not in the references section. In the sentence extracted from other manuscripts as the numbered references are of other studies?

p.14

Comment 14:

“Neither F11+ nor M11+ could significantly reduce the risk of groin and hip injuries.”

Can these stats be shown in a table?

Comment 15:

The paragraph below is very important, however the message is not clear. Please rephrase.

“Neither F11+ nor M11+ could significantly reduce the risk of groin and hip injuries. In addition to the reasons mentioned above, they could not mitigate hip and groin injuries since they provided no specific exercise for groin muscles. This appears to be a noticeable gap, as up to 12% of subelite football injuries occur in the groin region [41]. Wahlan et al. suggested that using Copenhagen adductor exercises may fill this gap and would be fit for inclusion in the FIFA11+ programme, as it requires no specific equipment and has three levels of difficulties [41]. The better efficacy of the M11+ in preventing hip injuries, however, might be due to increasing the volume of hip dynamic stretch exercise, namely, hip-in, in addition to manipulating how core area muscles are being trained. “

P.15

Comment 16:

Limitations

In fact the study did not recruit the expected number of teams to manage a randomized sample. It is not clear that the authors did recognize this limitation.

Comment 17:

Was the trial registered in a clinical trials platform?

6. PLOS authors have the option to publish the peer review history of their article (what does this mean?). If published, this will include your full peer review and any attached files.

Reviewer #1: No

Reviewer #2: No

---

## [Author Response · Author response to Decision Letter 0]

10 Aug 2022

Dear Editor,

Thank you so much for your great eyes to our work and highlighting those important points. We believe that the points you have raised would greatly enhance the quality of the manuscript. Thus, we tried to follow them in full and revise the manuscript accordingly. More specifically, regarding the data availability policy, we have now uploaded a minimized dataset as per your request. you will find the URL and DOI: 

URL:www.kaggle.com/dataset/1f3df0f9ccf87f9dbeaca960baf76527376189128948699c4dec7eb056098834

https://doi.org/10.34740/KAGGLE/DSV/4054705

All means and standard deviations have already been mentioned in the manuscript. The changes we made based on your comments are highlighted in red throughout the manuscript.

More importantly, regarding your following comment ‘’Viewing your abstract, we can see that it was previously published by Springer: "Al Attar, W.S.A., Soomro, N., Pappas, E. et al. How Effective are F-MARC Injury Prevention Programs for Soccer Players? A Systematic Review and Meta-Analysis. Sports Med 46, 205–217 (2016). https://doi.org/10.1007/s40279-015-0404-x" We believe that there must be a mistake simply because any part of this manuscript has not been published up to this point. Further, the link you referred belongs to a systematic review which was first appeared in the literature 6 years ago, whereas this project has only one year old.

Reviewer #1: Dear authors,

I believe they did a good job and deserve credit for that.

The study is well designed, the introduction is robust, the methods are well explained and the discussion supports the results.

However, the conclusion can and should be improved, and should be presented with practical applications for the coach.

Furthermore, I believe that the limitations should be inserted at the end of the discussion.

Response: Dear Reviewer, we thank you so much for your positive review and for the positive and useful comments. We revised the conclusion and added recommendations for practitioners and trainers. Additionally, we inserted the limitation at the end of the discussion. The changes we made based on your comments are highlighted in light brown. 

Reviewer #2: The proposed manuscript is very interesting and has a noteworthy objective. Congratulations to the authors.

My suggestions follow below.

Response: Dear Reviewer, we thank you very much for your positive feedback on our work and for the detailed, supportive and constructive comments. They are very helpful to improve the manuscript’s quality. Therefore, our effort was to integrate the comments throughout the manuscript accurately. The changes based on your comments are highlighted in light blue.

P.8

Comment 1:

Please rephase the “Background”, specifically “…the latter has potentially reduced compliance and

implementation of the programme”

Response: Thank you. We revised the background accordingly. 

Comment 2:

Keywords: I suggest not repeating the keyword that already exists in the title to enhance search results.

Response: Thank you very much. We removed the keywords that are already a part of the title 

Comment 3:

“static stretches” in the warm-up? Or did the authors mean “dynamic stretches”?

Response: Thank you very much for your great eyes to the important words. According to the reports of our research assistant who was visiting the CG, athletes indeed performed some static stretches within the warm-up phase, with each stretch lasting approximately 12 seconds. 

Comment 4:

Is the discussion the conclusion in the abstract?

Response: Thank you for your comment. We added a one-sentence conclusion to the abstract. 

P.9

Comment 4:

I suggest rephrasing the sentence “It aims to tackle modifiable injury risk factors such as

neuromuscular control, static and dynamic balance, and the hamstring/quadriceps strength

ratio[8].” as they are not risk factors. They can be considered “key elements of effective injury prevention programmes…”

Response: Thank you so much. We revised the sentence accordingly. 

Comment 5:

Please rephrase the sentence “The latter, along with barriers such as prolonged duration, concerns regarding some exercises, and lack of a link to football-specific targets[14, 18], has been discussed as a potential reason for low compliance and poor implementation of the programme[14, 19].”

Response: Thank you so much. We revised the sentence accordingly. 

Comment 6:

Please state the reference for the sentence “Although it is widely accepted that an appropriate warm-up programme should improve performance, more recent studies indicated that the 11+ programme does not improve players' performance acutely but may reduce sprinting and agility compared to a dynamic warm-up programme.”

Response: Thank you so much. We cited a suitable reference to support the sentence’s content.

P.10

Comment 7:

In the abstract the authors stated that the CG did not perform injury prevention studies, however in the methods section, the contrary is stated.

Response: Thank you so much for your great eyes to the manuscript. The phrase ‘’without any’’ specific injury prevention exercises… was missing. We rewrote the sentence. 

Comment 7:

Was static stretching truly used?

Response: Well, it was used and, to my knowledge, is still being used even in some developed countries and on a roughly large scale, unfortunately. 

P.11

Comment 8:

Please add further information of how the instructions to perform the 11+ and the M11+ were carried out.

Response: Thank you. We would be glad to present further details here and in the manuscript. First, we instructed the trainer and players on how to perform the programmes. Furthermore, we provided the 11+ group with the 11+manual (it was already translated in Persian by IF-MARC, Iran branch of FIFA Medical Assessment and Research Centre) and asked them to familiarize themselves with the 11+. In addition, we provided them with the YouTube link to watch out the 11+ exercises. For the M11+, we provided the trainers and the players with a series of photo slides including all exercises embedded in the programme. Meanwhile, to assure accuracy of the application of the 11+, a research assistant was responsible for visiting the training sessions of the M11+ and F11+ groups and checking the implementation of the warm-up modalities. 

Comment 9:

Data collection and supervision: How where the assistants “blinding each other”?

Response: Thank you so much for highlighting these important points. We applied that approach to minimize any potential risk of bias during data collection. Three research assistants were responsible for their subsequent groups while having no idea about the study aims, the interventions, occurred injuries of the other groups, etc. They were only in touch with the research group leader and remained anonymous across the project. 

Comment 10:

Why was the “M. iliopsoas strengthening” and the “single-leg standing with a heel-to-toe movement cycle” selected? Can the authors support these choices with evidence? It would be great as the M11+ group showed few injuries.

Response: Thank you very much for your detailed and beneficial comments. We are pleased to briefly explain it for your perusal:

The idea of strengthening M. iliopsoas muscles and single-leg standing with a heel-to-toe movement cycle and also modifying some core exercises comes from Janda’s approach, namely, the muscular chain, which is also known as upper/lower cross syndromes, as well as Shirley Sahrmann’s approach:

Izraelski, J., Assessment and treatment of muscle imbalance: The Janda approach. The Journal of the Canadian Chiropractic Association, 2012. 56(2): p. 158.

Sahrmann, S., D.C. Azevedo, and L. Van Dillen, Diagnosis and treatment of movement system impairment syndromes. Brazilian journal of physical therapy, 2017. 21(6): p. 391-399.

P.12

Comment 11:

A table with Injury incidence/1000 h and rate ratios for each group would enhance the manuscript.

Response: Thank you for your suggestion. We appreciate your support for enhancing the manuscript and considering them all. In this case, however, as we already have three tables in the manuscript, we taught that having another table could be too much. Therefore, we just added a row to table two, including IIC rates. 

Comment 12:

Further detail on injuries would also be of great interest for the study: (Fuller et al, 2006).

Response: Thank you. We added additional information. 

p.13

Comment 13:

The second manuscript from (Lopes et al.) is not in the references section. In the sentence extracted from other manuscripts as the numbered references are of other studies?

Response: Thank you so much for highlighting that mistake. Following your comment, we went through the reference list and found some errors by the meaning of false matching the numbers and references. Thus, all references were updated and matched accurately. 

p.14

Comment 14:

“Neither F11+ nor M11+ could significantly reduce the risk of groin and hip injuries.”

Can these stats be shown in a table?

Response: Thank you for your attention. Given the few injuries that occurred in the hip and groin, we considered them and reported all as thigh injuries.

Comment 15:

The paragraph below is very important, however the message is not clear. Please rephrase.

“Neither F11+ nor M11+ could significantly reduce the risk of groin and hip injuries. In addition to the reasons mentioned above, they could not mitigate hip and groin injuries since they provided no specific exercise for groin muscles. This appears to be a noticeable gap, as up to 12% of subelite football injuries occur in the groin region [41]. Wahlan et al. suggested that using Copenhagen adductor exercises may fill this gap and would be fit for inclusion in the FIFA11+ programme, as it requires no specific equipment and has three levels of difficulties [41]. The better efficacy of the M11+ in preventing hip injuries, however, might be due to increasing the volume of hip dynamic stretch exercise, namely, hip-in, in addition to manipulating how core area muscles are being trained.

Response: Thank you for your comment. We revised the paragraph and hope that the readers now can clearly follow the argumentation and get the message.

P.15

Comment 16:

Limitations

In fact the study did not recruit the expected number of teams to manage a randomized sample. It is not clear that the authors did recognize this limitation.

Response: Thank you. We revised the limitation and pointed out the lack of a recruiting sample in addition to the short monitoring period. 

Comment 17:

Was the trial registered in a clinical trials platform?

Response: Thank you. It was not registered

---

## [Decision Letter · Decision Letter 1]

12 Sep 2022

PONE-D-22-14375R1Effects of the FIFA 11+ and a modified warm-up programme on injury prevention and performance improvement among youth male football players.PLOS ONE

Dear Dr. Asgari,

Thank you for submitting your manuscript to PLOS ONE. After careful consideration, we feel that it has merit but does not fully meet PLOS ONE’s publication criteria as it currently stands. Therefore, we invite you to submit a revised version of the manuscript that addresses the points raised during the review process.

Dear authors,

The authors made a great work in improving the work by following suggestion made by both reviewers. Reviewer 1 already recommend acception. Even so, I agree with the opinion of the second reviewers and for that reason, I suggest another round to improve the topic about static stretching being considered a good strategy for injury prevention.

Thank you

We look forward to receiving your revised manuscript.

Kind regards,

Rafael Franco Soares Oliveira

Academic Editor

PLOS ONE

Journal Requirements:

Additional Editor Comments:

Dear authors,

The authors made a great work in improving the work by following suggestion made by both reviewers. Reviewer 1 already recommend acception. Even so, I agree with the opinion of the second reviewers and for that reason, I suggest another round to improve the topic about static stretching being considered a good strategy for injury prevention.

Thank you

Reviewers' comments:

Reviewer's Responses to Questions

**Comments to the Author**

1. If the authors have adequately addressed your comments raised in a previous round of review and you feel that this manuscript is now acceptable for publication, you may indicate that here to bypass the “Comments to the Author” section, enter your conflict of interest statement in the “Confidential to Editor” section, and submit your "Accept" recommendation.

Reviewer #1: All comments have been addressed

Reviewer #2: All comments have been addressed

2. Is the manuscript technically sound, and do the data support the conclusions?

Reviewer #1: Yes

Reviewer #2: Yes

3. Has the statistical analysis been performed appropriately and rigorously? 

Reviewer #1: Yes

Reviewer #2: I Don't Know

4. Have the authors made all data underlying the findings in their manuscript fully available?

Reviewer #1: Yes

Reviewer #2: Yes

5. Is the manuscript presented in an intelligible fashion and written in standard English?

Reviewer #1: Yes

Reviewer #2: Yes

6. Review Comments to the Author

Reviewer #1: Dear author, thank you for submitting the manuscript.

I believe that the authors did a good job of reformulating the manuscript according to the reviewers' requests. In that sense, I have no further reservations about the manuscript.

Reviewer #2: I thank the authors for their great effort in replying adequately to all the suggestion made. I have one final concern. I recommend a like a deeper discussion around the topic “static stretching” to be reflected in this manuscript. We understand that static stretching is not recommended to initiate sports practice. It reduces strength. How can the authors defend this modification to widely accepted injury prevention program. Please state this clearly in the manuscript.

7. PLOS authors have the option to publish the peer review history of their article (what does this mean?). If published, this will include your full peer review and any attached files.

Reviewer #1: No

Reviewer #2: No

---

## [Author Response · Author response to Decision Letter 1]

16 Sep 2022

Dear respected editor/ reviewers

On behalf of our research group, I do thank you so much again for your positive reviews and commentary on our work. We do believe that the comments you made have widely improved the quality of our paper, so we highly appreciate your efforts. Just to reflect well to the last comment provided by the second reviewer which is a kind of misunderstanding/miscommunication, I do refer to the first revision round where I clearly stated that based on the research assistant, the Control Group (CG) used some static stretches as part of their routine warm up before main training. So, there was no any kind of static stretches neither in 11+ nor in the modified 11+ and as you accurately mentioned, it has been scientifically proven that static stretches must not be used for warming up. 

To eliminate such misunderstanding, bellow you’ll find the original comments and the responses from the first revision round: 

Comment 3:

“static stretches” in the warm-up? Or did the authors mean “dynamic stretches”?

Response: Thank you very much for your great eyes to the important words. According to the reports of our research assistant who was visiting the CG, athletes indeed performed some static stretches within the warm-up phase, with each stretch lasting approximately 12 seconds. 

Comment 7:

Was static stretching truly used?

Response: Well, it was used and, to my knowledge, is still being used even in some developed countries and on a roughly large scale, unfortunately. 

Given that there is no other comments to be reflected, I assume that this clarification suffices and therefore, we did not provide any changes on the manuscript.

---

## [Editor Report · Decision Letter 2]

19 Sep 2022

Effects of the FIFA 11+ and a modified warm-up programme on injury prevention and performance improvement among youth male football players.

PONE-D-22-14375R2

Dear Dr. Asgari,

We’re pleased to inform you that your manuscript has been judged scientifically suitable for publication and will be formally accepted for publication once it meets all outstanding technical requirements.

Kind regards,

Rafael Franco Soares Oliveira

Academic Editor

PLOS ONE

Additional Editor Comments (optional):

Dear authors,

Congratulations on your work! After the clarification of the authors on the previous comments, I believe that this work can be accepted for publication.

Thank you.

Best regards
---

## [Editor Report · Acceptance letter]

22 Sep 2022

PONE-D-22-14375R2 

Effects of the FIFA 11+ and a modified warm-up programme on injury prevention and performance improvement among youth male football players. 

Dear Dr. Asgari:

I'm pleased to inform you that your manuscript has been deemed suitable for publication in PLOS ONE. Congratulations! Your manuscript is now with our production department. 

Kind regards, 

on behalf of

Dr. Rafael Franco Soares Oliveira 

Academic Editor

PLOS ONE